# COVID-19 Is a Confounder of Increased *Candida* Airway Colonisation

**DOI:** 10.3390/pathogens12030463

**Published:** 2023-03-15

**Authors:** Margaux Froidefond, Jacques Sevestre, Hervé Chaudet, Stéphane Ranque

**Affiliations:** 1Faculté des Sciences Médicales et Paramédicales, Aix Marseille University, IRD, AP-HM, SSA, VITROME, 13005 Marseille, France; 2IHU Méditerranée Infection, 19-21 Boulevard Jean Moulin, 13005 Marseille, France

**Keywords:** *Candida* colonisation, *Candida* airway colonisation, SARS-CoV-2 infection, fungemia, risk factors, case-control study

## Abstract

An increased incidence of invasive fungal infection was reported in SARS-CoV-2-infected patients hospitalised in the intensive care unit. However, the impact of COVID-19 on *Candida* airway colonisation has not yet been assessed. This study aimed to test the impact of several factors on *Candida* airway colonisation, including SARS-CoV-2 infection. We conducted a two-pronged monocentric retrospective study. First, we analysed the prevalence of positive yeast culture in respiratory samples obtained from 23 departments of the University Hospital of Marseille between 1 January 2018 and 31 March 2022. We then conducted a case-control study, comparing patients with documented *Candida* airway colonisation to two control groups. We observed an increase in the prevalence of yeast isolation over the study period. The case-control study included 300 patients. In the multivariate logistic regression, diabetes, mechanical ventilation, length of stay in the hospital, invasive fungal disease, and the use of antibacterials were independently associated with *Candida* airway colonisation. The association of SARS-CoV-2 infection with an increased risk of *Candida* airway colonisation is likely to be a consequence of confounding factors. Nevertheless, we found the length of stay in the hospital, mechanical ventilation, diabetes, and the use of antibacterials to be statistically significant independent risk factors of *Candida* airway colonisation.

## 1. Introduction

In December 2019, the first case of SARS-CoV-2 infection was described in Wuhan, China, heralding the emergence of a global pandemic responsible for 546 million cases and 6.3 million deaths [1,2]. The morbidity associated with COVID-19 is heterogeneous, ranging from asymptomatic infections to acute respiratory distress syndrome (ARDS), requiring urgent hospitalisation in an intensive care unit (ICU) [3]. ICU hospitalisation and subsequent invasive therapies may generate infectious complications, potentially associated with significant morbidity and mortality [4]. 

Yeast isolation from respiratory samples is common, notably in ICU patients receiving mechanical ventilation [5,6]. Yeast colonisation has been reported in 30% of patients receiving ventilation over 48 h and in up to 50% of patients diagnosed with ventilator-associated pneumonia (VAP) [7]. Similarly, recent studies report such respiratory colonisation in up to 21% of patients with COVID-19 [8]. The known risk factors for *Candida* airway colonisation include exposure to broad-spectrum antibiotics, the use of corticosteroids and immunosuppressants, the use of mechanical ventilation, neutropaenia, and diabetes [9,10]. Some studies have also reported that the duration of mechanical ventilation and the length of stay in the ICU are associated with a poor clinical outcome [11,12]. However, data regarding the impact of SARS-CoV-2 infection on respiratory tract colonisation by *Candida* yeast are scarce. This study aimed to assess the association of risk factors, particularly SARS-CoV-2 infection, with *Candida* respiratory tract colonisation. A secondary objective was to assess the effect of systemic antibiotic treatments on *Candida* airway colonisation.

## 2. Methods

### 2.1. Study Design

We conducted a two-pronged, single-centre retrospective study. The first approach involved a longitudinal observational study based on the “Méditerranée Infection Data Warehousing and Surveillance” (MIDAS) database. This programme, hosted at the Institut Hospitalo-Universitaire (IHU) Méditerranée Infection, combines syndromic and conventional surveillance and collects data from all the samples received at the microbiology laboratory, any subsequent tests performed, and the results obtained, on a weekly basis [13]. We analysed the prevalence of yeast isolation in respiratory samples from patients hospitalised in 21 units at the Assistance Publique—Hôpitaux de Marseille (AP-HM) university hospital of Marseille (France) from 1 January 2018 to 31 March 2022. The second approach was a case-control study comparing patients with *Candida* airway colonisation to two groups of control patients, who had no positive respiratory sample yeast culture. 

### 2.2. Inclusion Criteria

The patients included in the study between 1 April 2021 and 30 November 2021 were required to meet the following inclusion criteria: 

Be over 18 years old.

Be hospitalised in the AP-HM during the study period, with at least three respiratory samples obtained for the culture.

### 2.3. Exclusion Criteria

Patients under 18 years of age, patients without respiratory samples, or patients who objected to the use of personal data for research purposes were excluded from the analysis.

### 2.4. Case Definition

*Candida* airway colonisation cases were defined by three or more respiratory samples growing yeasts on culture and obtained on separate days.

### 2.5. Control Definitions

The patients included in the control groups had to meet the following criteria:

Absence of yeast cultured from at least three respiratory samples obtained on separate days.

The positive control (BactC) group included patients with a positive bacterial culture (*Haemophilus influenzae, Klebsiella pneumoniae, Streptococcus pneumoniae, Pseudomonas aeruginosa...)* in a respiratory sample.

The negative control (NegC) group included patients for whom respiratory sample cultures were negative, i.e., neither bacteria nor yeast were cultured. 

### 2.6. Data Collection 

In the first part of the study, the respiratory sample culture results from 21 units of the AP-HM were collected through the MIDAS programme. In the second part of the study, clinical data were collected through the hospital computerised medical records. The demographic information collected included: sex, age, medical history (obesity, high blood pressure (HBP), heart disease, diabetes, smoking, chronic respiratory failure, chronic renal failure with a global filtration rate (GFR) < 60 mL/min/1.73 m^2^), a history of haematological disease, cancer, a history of transplantation, immunosuppression, and the use of immunosuppressive therapy (long-term corticosteroid therapy, cancer chemotherapy, or anti-rejection therapy for transplant patients).

The data collected on the hospital stay included: length of stay, hospitalisation in the ICU or medical ward, diagnosis of SARS-CoV-2 infection with identification of the variant, use of oxygen therapy and delivery technique (low-dose oxygen therapy, high-flow nasal cannula (HFNC), non-invasive ventilation (NIV), mechanical ventilation (MV)), use of extracorporeal membrane oxygenation (ECMO), use of enteral or parenteral nutrition, abdominal surgery during hospitalisation, the administration of antibacterial, antiviral, or antifungal antibiotics, diagnosis of invasive fungal disease during the stay, detection of respiratory viruses, the type of respiratory samples obtained (bronchoalveolar lavage (BAL), tracheobronchial aspiration (TBA), sputum cytobacteriological examination (CBES)), the use of immunomodulatory therapies (notably, dexamethasone, methylprednisolone, and tocilizumab), and the patient’s outcome (deceased or alive). The time from admission to isolation of the first yeast in a respiratory sample was noted in patients in the case group. Similarly, for the BactC group, the time between admission and the first positive bacterial culture of respiratory samples was recorded.

### 2.7. Statistical Analysis 

The case-control study data were analysed using SAS 9.2 for Windows (SAS Institute Inc., Cary, NC, USA). Continuous variables were expressed as the mean (SD), while categorical variables were expressed as proportions and percentages. Continuous variables were compared using ANOVA. Categorical variables were compared using the Chi square or Fisher’s exact tests, as required. All statistical tests were two-sided, with a *p* < 0.05 significance level. Univariate and multivariate unconditional logistic regression analyses were performed to estimate odds ratios (ORs) with a 95% confidence interval (CI). All covariates with a *p* < 0.20 significance level in the univariate analysis were included in the multivariate analyses. Three multivariate logistical regression models were computed: the case group was compared either to (i) all controls (including both the NegC and BactC groups), (ii) only the BactC, or (iii) only the NegC groups. For each control group model, a stepwise selection was performed to retain the most parsimonious model, including the covariates that displayed an independent statistically significant effect on the risk of airway fungal colonisation.

## 3. Results 

### 3.1. Time Trends in the Prevalence of Yeast-Positive Culture in Respiratory Samples

Between 1 January 2018 and 31 March 2022, a total of 17,408 respiratory samples were collected, of which 12,881 grew bacteria and/or fungi in culture. Most (17,102 (98.2%)) of the samples collected over the study period originated from ICU patients. Yeast was isolated in 3593 samples, amounting to 28% of the positive cultures. The distribution of yeast species during this period is described in Figure 1. 

Of those samples, 2658 (74%) grew *C. albicans*, 272 (8%) grew *C. tropicalis,* 217 (6%) grew *C. glabrata*, 123 (3%) grew *C. dubliniensis*, and 115 (3%) grew *C. parapsilosis*. The prevalence of yeast isolation increased from February 2020, concomitantly with the onset of the COVID-19 epidemic in Marseille, with a predominance of *C. albicans* (17% of yeasts isolated). In April 2020, up to 23% of culture samples grew *C. albicans.* This trend continued over the following year, with *C. albicans* being isolated from 24% of respiratory samples in May 2021. A relative decrease in the prevalence of *C. albicans* occurred in July 2021 (16%), August 2021 (17%), and November 2021 (14%). Finally, in 2022, 20% of the samples were positive for *C. albicans* in January, 22% in February, and 19% in March. This time trend is illustrated in Figure 2 and Figure 3.

### 3.2. Patient Demographics

Between 1 April 2021 and 30 November 2021, 300 patients (100 in each group) were included in the case-control study (Table 1). The three groups were homogeneous regarding age (*p =* 0.8752) and sex (*p =* 0.7605). A total of 80 patients (80%) in the case group were admitted to the ICU, compared to 62 (62%) and 18 (18%) patients in the BactC and NegC groups, respectively (*p* < 10^−4^). Comorbidities were homogenously distributed among the three groups. Of the 79 patients with a history of chronic respiratory failure, 32 had chronic obstructive pulmonary disease (COPD), 18 had cystic fibrosis, 8 had diffuse interstitial lung disease, 7 had pulmonary fibrosis, 6 had bronchial dilatation, 6 had emphysema, 1 had severe asthma, and 1 had respiratory failure consecutive to pulmonary tuberculosis. Diabetes was statistically significantly (*p* < 10^−4^) more frequent in the case group (31%) compared to the BactC and NegC groups—26% and 7%, respectively. In contrast, solid malignancies were rarer (*p* = 0.0114) in the case group (7%) compared to the BactC and NegC groups—16% and 22%, respectively. Similarly, immunosuppression was rarer (*p* = 0.0212) in the case group (17%) compared to the BactC and NegC groups—31% and 33%, respectively. Among immunocompromised patients, 42 had a history of organ transplantation, including 33 lung transplant patients, 16 had a history of chronic inflammatory or autoimmune disease, 12 were receiving cancer chemotherapy, 7 had been treated for a haematological disease, and 4 were infected with HIV. 

### 3.3. Characteristics of Hospital Stay 

The use of parenteral nutrition (*p =* 0.1697) and oxygen delivery devices, including low-dose oxygen (*p* = 0.2129), HNFC (*p* = 0.6004), and non-invasive ventilation (*p* = 0.2529), was homogenously distributed within the three groups. In contrast to BAL (*p =* 0.8788) and CBES (*p =* 0.9505), TBA was statistically significantly most often performed in the case group (*p <* 10^−4^).

Regarding the factors associated with the ICU, MV (*p <* 10^−4^), ECMO (*p <* 10^−4^), central venous catheter (*p <* 10^−4^), abdominal surgery during hospitalisation (*p =* 0.0323), and parenteral nutrition implementation (*p <* 10^−4^) were statistically significantly most often observed in the case group patients. Similarly, the mean hospital stay was longer (*p <* 10^−4^) in the case group, with 46 (±37) days compared to 26 (±22) and 9 (± 8) days for the BactC and NegC groups, respectively. Case-fatality rates were also higher (*p* = 0.0032), with 29 (29%) fatal outcomes in the case group compared to 23 (23%) and 10 (10%) in the BactC and NegC groups, respectively.

### 3.4. Microbiological Characteristics 

A total of 53 (53%) patients in the case group had SARS-Co-V-2 infection, compared with 29% and 17% in the BactC and NegC groups, respectively (*p <* 10^−4^). The mean timescale between hospital admission and the first culture-positive respiratory sample was seven days in both the Case and BactC patients. Within the *Candida* genus (Figure 4), *C. albicans* (57%) was predominantly isolated, followed by *C. glabrata* (17%), C. *dubliniensis* (7%), *Kluyveromyces marxianus* (5%)*,* and *C. tropicalis* (4%). 

Of the 91 respiratory samples with a positive bacterial culture in the case group patients, the most frequent bacterial coinfection involved Enterobacteriaceae (47%), including *Klebsiella pneumoniae* (18%), followed by *Pseudomonas aeruginosa* (19%) and methicillin-susceptible *Staphylococcus aureus* (11%) (Figure 5). In the BactC group patients, 138 respiratory specimens yielded Enterobacteriaceae (40%), including *E. coli* (11%) and *Klebsiella* sp. (9%), 28% yielded *P. aeruginosa*, and 17% yielded methicillin-susceptible *S. aureus*. The distribution of bacterial species is illustrated in Appendix A. The use of antibacterial therapy was significantly more frequent (*p <* 10^−4^) in the case and BactC patients (88% each) compared to those in the NegC group (35%)**.**

A viral respiratory infection was more frequently diagnosed (*p <* 10^−4^) in patients in the case group (25%) compared to patients in the BactC (13%) and NegC (1%) groups. Antiviral treatment was initiated during the hospital stay in 27 (27%) patients in the case group compared to 12% and 1% in the BactC and the NegC groups, respectively (*p <* 10^−4^).

Invasive fungal disease (IFD) occurred in 16 patients (16%) in the case group (five candidaemia, ten pulmonary aspergillosis, and one *C. albicans* pleurisy). In the BactC group, fungal infection was documented in three patients (one candidaemia and two pulmonary aspergilloses). No fungal infections were documented in the NegC group (*p <* 10^−4^). Twenty-six (26%) patients in the case group received systemic antifungal therapy, compared with 5% and 2% in the BactC and the NegC groups, respectively (*p <* 10^−4^). Notably, two patients in the NegC group received pre-emptive antifungal therapy.

### 3.5. Description of Candidaemia Cases

The clinical characteristics and microbiological findings of the five case group patients who developed candidaemia during their ICU stay are presented in Table 2. Comparatively, one patient from the BactC group developed *C. parapsilosis* candidaemia. This patient was 53 years old, without comorbidities at admission, and had been hospitalised for 46 days in the ICU for polytrauma management. He was intubated, had no documented SARS-CoV-2 infection, and had not had abdominal surgery. No *Candida* colonisation had been documented in previous samples. The initial caspofungin antifungal treatment was then switched to fluconazole. Additionally, a documented *Enterobacter cloacae* and *K. pneumoniae* VAP was treated with piperacillin-tazobactam.

### 3.6. Documented SARS-CoV-2 Infection

A SARS-Co-V-2 infection was diagnosed in 53 (53%) patients in the case group, compared to 29% and 17% in the BactC and NegC groups, respectively. The Alpha variant was the most frequently involved. It was documented in 26 patients from the case group, 9 patients from the BactC group, and 10 patients from the NegC group. The Delta variant was detected in 13 patients in the case group, 9 patients in the BactC group, and 1 patient in the NegC group (Appendix A). The Alpha variant was predominant during April and May, whereas the Delta variant became predominant from July 2021. The monthly distribution of variants between April and November 2021 is shown in Appendix A.

The immunomodulatory treatments used against SARS-CoV-2 are summarised in Table 3. Dexamethasone treatment was more frequently (*p <* 10^−4^) used in the case group patients (53%), compared with 28% and 10% in the BactC and NegC groups, respectively. Methylprednisolone was initiated in 37%, compared with 16% and 3% in the BactC and NegC groups, respectively (*p <* 10^−4^). Tocilizumab was homogenously (*p =* 0.6045) prescribed in the case (6%) and BactC (7%) groups.

### 3.7. Exposure to Antibacterials

The exposure to antibacterials in each patient group is summarised in Appendix A. Piperacillin-tazobactam was the most frequent antibacterial used. It was administered in 44% of patients in the case group, compared with 34% and 14% in the BactC and NegC groups, respectively. Carbapenem was administered in 35% of the case group patients. In the BactC group patients, cefotaxime, ceftriaxone (23%), and group A penicillins (23%) were the most used after piperacillin-tazobactam. Penicillin A was administered in 13% of the NegC group patients.

### 3.8. Multivariate Analysis

The results of the multivariate analyses are summarised in Table 4. The first analysis, comparing the case group and both control groups (BactC+NegC), found a statistically significant association between *Candida* airway colonisation and diabetes, mechanical ventilation, length of hospital stay, the development of invasive fungal disease, and the use of some antibacterial antibiotics, including carbapenems, cefepime, and piperacillin/tazobactam. When compared to the BactC group, *Candida* airway colonisation was statistically significantly associated with mechanical ventilation, diabetes, length of hospital stay, and the use of cefepime and carbapenems. In contrast, a history of solid malignancy appeared to be a protective factor. When compared to the NegC group, *Candida* airway colonisation was statistically significantly associated with the length of hospitalisation, antifungal treatment, and linezolid administration. 

## 4. Discussion

Our first finding was a dramatic increase in the number of respiratory samples analysed at the AP-HM university hospital microbiology laboratory since March 2020, in comparison with the two previous years. Most of the respiratory samples originated from ICU patients, and the prevalence of positive yeast (mainly *Candida* spp.) culture increased in parallel. This increased prevalence of *Candida* airway colonisation was concomitant to the SARS-CoV-2 epidemic and roughly correlated with the successive epidemic waves in France, notably in March and April 2020, the last quarter of 2020, from February to May 2021, in August and September 2021, and from December 2021 to January 2022 [14]. One explanation for this increased number of respiratory samples is probably the rising number of patients admitted to the ICU due to severe SARS-CoV-2 infection since March 2020. Some 106,000 patients with COVID-19 were admitted to ICUs between 1 March 2020 and 30 June 2021 in France. Comparatively, 19,000 patients were admitted to French ICUs due to severe influenza between 2014 and 2019 [15]. The documentation of possible respiratory coinfections in ICU patients is probably the main reason for this increase in respiratory samples. Most patients admitted to the ICU received mechanical ventilation and were thus exposed to VAP [8].

The distribution of yeast species, *C. albicans* (57%) and *C. glabrata* (17%), in the respiratory sample cultures that we observed was in line with previous reports [5]. Similarly, the seven-day timeframe between admission and the first positive *Candida* culture that we observed was comparable to the 6 ± 1.6-day timeframe reported by Hedderwick et al. in 16 ICU patients [16]. 

Several studies have assessed the risk of candidaemia in SARS-CoV-2-infected patients [17,18,19,20,21]. We tested whether SARS-CoV-2 infection was an independent factor of *Candida* airway colonisation. In our univariate analysis, SARS-CoV-2 infection was statistically significantly more frequently diagnosed in patients with *Candida* spp. airway colonisation. However, SARS-CoV-2 infection is likely to be a confounding factor, since its independent effect was not statistically significant in the multivariate analysis. Likewise, patients with *Candida* spp. airway colonisation were significantly more frequently hospitalised in the ICU, which is in line with previous studies that reported 50% to 86% *Candida* colonisation rates in patients who had a prolonged stay in the ICU [22,23,24,25]. However, this association was not statistically significant in the multivariate analysis, which highlighted ICU-related factors such as mechanical ventilation as independent risk factors of *Candida* spp. airway colonisation. 

In the univariate analysis, diabetes, mechanical ventilation, ECMO, central venous catheters, abdominal surgery, length of hospitalisation, and antibacterial therapy were associated with colonisation. A fatal outcome occurred more frequently in colonised patients. However, the multivariate analysis only found diabetes, length of hospitalisation, mechanical ventilation, and antibacterial therapy to be associated with *Candida* colonisation.

Regarding mechanical ventilation, Arastehfar et al. reported respiratory *Candida* colonisation in 20% of patients after 48 h of mechanical ventilation [18]. In line with our findings, several studies reported the association of a protracted hospital stay and mechanical ventilation with *Candida* airway colonisation [5,19,26]. Keeping with our findings, Chakraborti et al. and Erami et al. identified diabetes as a risk factor for *Candida* airway colonisation [10,27]. However, in contrast to Erami et al., who identified solid malignancies, chronic renal failure, and pre-existing cardiovascular diseases as *Candida* airway colonisation risk factors, none of these factors were significant in our study [10]. In contrast, we found that a solid malignancy history protected from *Candida* airway colonisation. This debatable result could be explained by the heterogeneity of the patient population in each study. The predominance of ICU patients in our study represents a recruitment bias because patients with advanced cancer are generally not admitted to the ICU. The role of chronic renal failure remains a matter of debate [16]. 

In our study, the use of antibacterials was statistically significantly associated with an increased risk of *Candida* airway colonisation. Subgroup analyses identified some antibacterials that were more frequently associated with a risk of colonisation—notably, carbapenems and cefepime. These results are aligned with those of Delisle et al. and Charles et al., who reported an increased use of antibacterials in *Candida*-colonised patients [12,28]. Notably, these two studies reported on broad-spectrum antibacterials but did not specify which antibacterial classes were used.

The development of invasive fungal disease and the use of antifungal therapy were associated with *Candida* colonisation when compared to non-colonised (NegC) patients with culture-negative respiratory specimens. One explanation might be that patients in the NegC group were less severe and less likely to be treated with systemic antifungals than ICU patients. Five fungaemia occurred in the group of patients colonised with *Candida* (5%), whereas only one occurred in the non-colonised patient group. These data are in line with several previous studies [29]. Moreover, the relevance of assessing *Candida* colonisation in patients at risk of invasive candidiasis has been demonstrated by Pittet et al. [22].

The use of high-dose corticosteroids (dexamethasone and methylprednisolone) was not shown to be an independent risk factor in our multivariate analysis. Indeed, its effect on *Candida* colonisation remains under debate [16]. Likewise, in contrast to other studies, we did not find that tocilizumab was an independent risk factor for *Candida* colonisation, which might be due to the relatively small number of patients treated with it [23,24].

Our multivariate analysis did not show a difference in mortality between the colonised and non-colonised groups. It should be noted that the association between *Candida* colonisation and ICU patients’ mortality is disputed, with conflicting reports in the literature [12]. 

Among the colonised patients in our study, only one had a documented *Candida* lung infection. *Candida* pneumonia is a rare infection. In their study on 25 autopsied patients, El-Biary et al. reported 10 who were colonised with *Candida,* only 2 of whom had histologically documented pneumonia [30]. Similarly, among 232 autopsied ICU patients, Meersseman et al. reported histologically documented pneumonia in 135 (58%). Of these, 77 (57%) had *Candida* spp. isolated from *pre-mortem* respiratory samples, and none displayed histological features of *Candida* pneumonia.

The limitations of our study include its monocentric design, which means that our results can only be extrapolated to other settings with caution. Retrospective studies are also known to be prone to bias. Nevertheless, our findings are strengthened by our choice to define colonisation through the positive culture of three distinct respiratory samples, whereas other studies define colonisation on the basis of a single positive sample [29,31]. This is likely to increase the robustness of our findings. Furthermore, our choice to use distinct patient control groups allowed us to highlight robust risk factors, which have an impact on a fairly heterogeneous group of patients.

## 5. Conclusions

Both the incidence and prevalence of *Candida*-positive respiratory samples increased in parallel to the SARS-CoV-2 epidemic. However, SARS-CoV-2 infection is likely to be a confounding factor rather than an independent risk factor. While SARS-CoV-2 infection was statistically significantly more frequently diagnosed in patients with *Candida* airway colonisation in our univariate analysis, the length of hospitalisation, the use of mechanical ventilation, diabetes, and the use of antibacterial antibiotics, but not SARS-CoV-2 infection, were independently associated with *Candida* airway colonisation in our multivariate analysis. Notably, respiratory *Candida* colonisation was significantly associated with a sevenfold increase in the risk of developing invasive fungal disease—in particular, fungaemia. 

## Figures and Tables

**Figure 1 pathogens-12-00463-f001:**
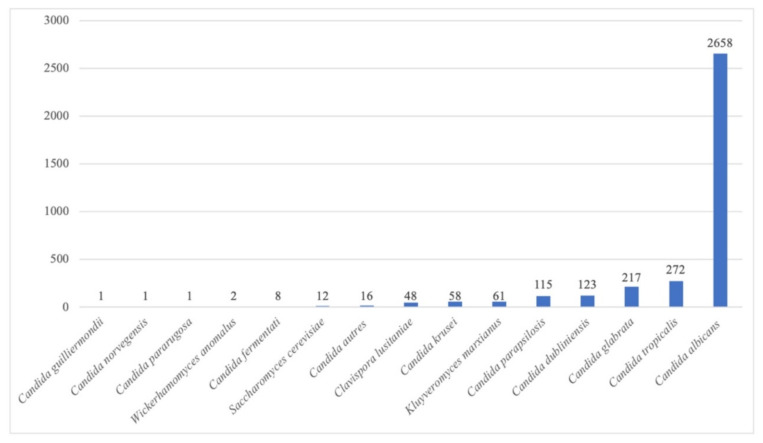
Distribution of yeast species cultured from respiratory samples taken at the university hospital of Marseille (AP-HM) from 1 January 2018 to 31 March 2022.

**Figure 2 pathogens-12-00463-f002:**
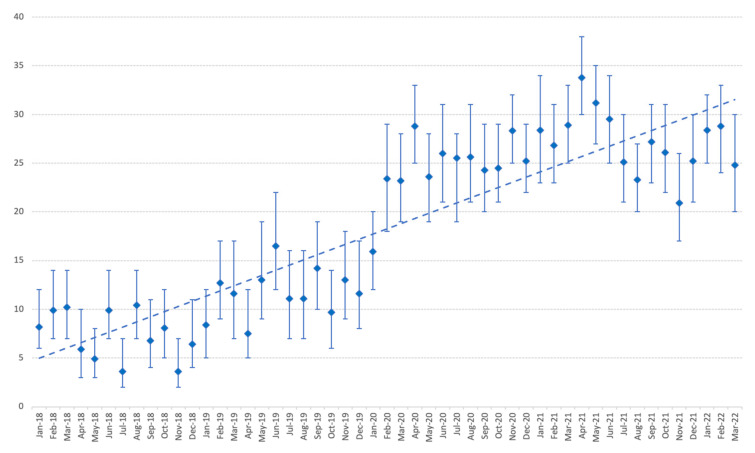
Evolution of the yeast rates (and 95% confidence intervals) among culture-positive respiratory specimens at the university hospital of Marseille (AP-HM), from 1 January 2018 to 31 March 2022.

**Figure 3 pathogens-12-00463-f003:**
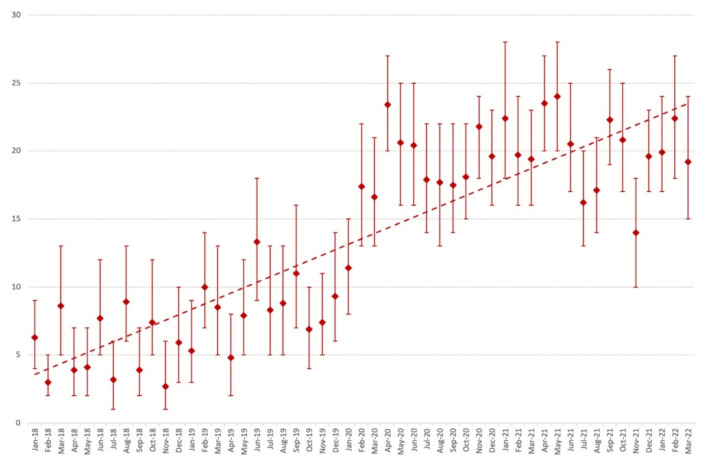
Evolution of *C. albicans* rates (and 95% confidence intervals) among culture-positive respiratory specimens from 1 January 2018 to 31 March 2022.

**Figure 4 pathogens-12-00463-f004:**
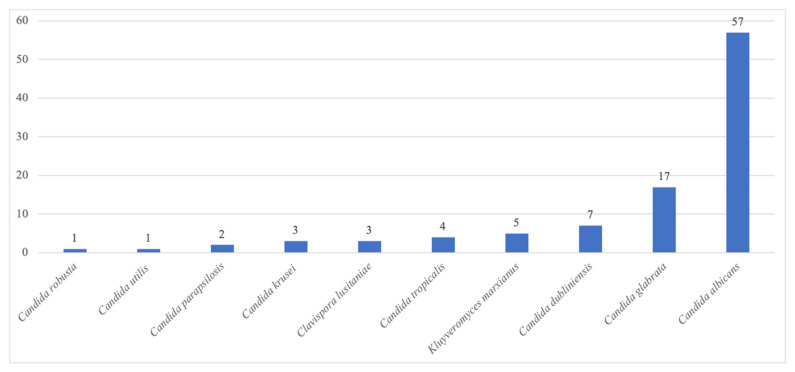
Distribution of the *Candida* species cultured from respiratory samples taken from the *Candida* airway colonisation group from 1 April to 30 November 2021 at the university hospital of Marseille (AP-HM).

**Figure 5 pathogens-12-00463-f005:**
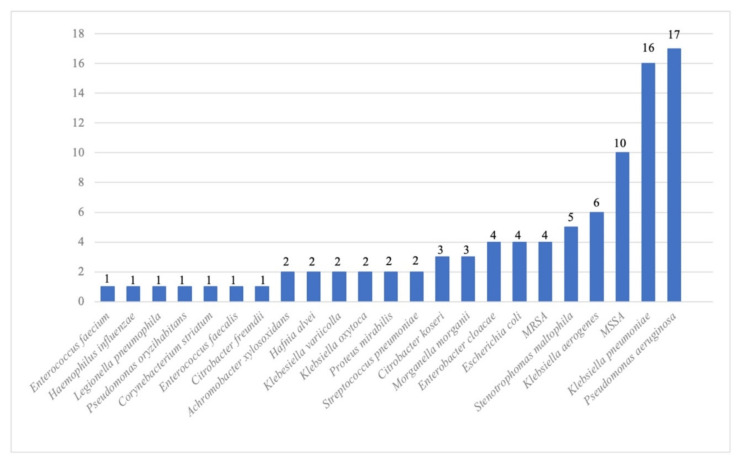
Distribution of the bacterial species cultured from respiratory samples in the *Candida* airway colonised patients’ group from 1 April to 31 November 2021.

**Table 1 pathogens-12-00463-t001:** Characteristics of the 300 patients included in the case, BactC, and NegC control groups between 1 April and 30 November 2021 at AP-HM.

n (%)	Case (n = 100)	Bbact (n = 100)	NegC (n = 100)	*p* ***
Demographic characteristics				
Males	66 (66)	64 (64)	61 (61)	0.7605
Age, years ^†^	57 ± 16	58 ± 16	58 ± 15	0.8752
Intensive care unit	80 (80)	62 (62)	18 (18)	**<10^−4^**
Hospital length of stay, days ^†^	46 ± 37	26 ± 22	9 ± 8	**<0.0001**
Comorbidities				
BMI				
BMI < 25	33 (44)	46 (56)	51 (59)	0.1283
25 < BMI < 30	19 (25)	21(25)	17 (20)	0.6033
30 ≤ BMI < 35	16 (21)	9 (11)	10 (12)	0.1200
35 ≤ BMI < 40	7 (9)	3 (4)	4 (5)	0.2691
40 ≥ BMI	0 (0)	3 (4)	2 (2)	0.2659
High blood pressure	42 (42)	39 (39)	31 (31)	0.2511
Cardiovascular disease	15 (15)	20 (20)	14 (14)	0.4695
Diabetes	31 (31)	26 (26)	7 (7)	**<10^−4^**
Smoking	25 (25)	21 (21)	28 (28)	0.5149
Chronic respiratory failure	28 (28)	28 (28)	23 (23)	0.6508
Chronic renal failure ^‡^	10 (10)	15 (15)	7 (7)	0.1801
Solid cancer	7 (7)	16 (16)	22 (22)	**0.0114**
Haematological disease	0 (0)	3 (3)	4 (4)	0.1493
Transplantation	11 (11)	19 (19)	14 (14)	0.2712
Immunosuppression	17 (17)	31 (31)	33 (33)	**0.0212**
Immunosuppressive treatments ^§^	15 (15)	28 (28)	26 (26)	0.0629
Ventilation/oxygen therapy				
Low-dose oxygen therapy	6 (6)	13 (13)	12 (12)	0.2129
HFNC	1 (1)	2 (2)	3 (3)	0.6004
Non-invasive ventilation	1 (1)	5 (5)	3 (3)	0.2529
Mechanical ventilation	80 (80)	56 (56)	15 (15)	**<10^−4^**
ECMO	22 (22)	13 (13)	1 (1)	**<10^−4^**
Nutrition	77 (77)	53 (53)	14 (14)	
Enteral nutrition	76 (76)	50 (50)	14 (14)	**<10^−4^**
Parenteral nutrition	1 (1)	3 (3)	0 (0)	0.1697
Central venous catheter	79 (79)	58 (58)	15 (15)	**<10^−4^**
Abdominal surgery	10 (10)	2 (2)	4 (4)	**0.0323**
SARS-CoV-2 infection	53 (53)	29 (29)	17 (17)	**<10^−4^**
Antibiotic therapy	88 (88)	88 (88)	35 (35)	**<10^−4^**
Antifungals	26 (26)	5 (5)	2 (2)	**<10^−4^**
Antivirals	27 (27)	12 (12)	1 (1)	**<10^−4^**
Nature of respiratory sample				
BAL	78 (78)	76 (76)	75 (75)	0.8788
TBA	70 (70)	43 (43)	12 (12)	**<10^−4^**
CBES	27 (27)	26 (26)	28 (28)	0.9505
Respiratory viral infection	25 (25)	13 (13)	1 (1)	**<10^−4^**
Fungal infection	16 (16)	3 (3)	0 (0)	**<10^−4^**
Fatality, n (%)	29 (29)	23 (23)	10 (10)	**0.0032**

* Statistically significant differences appear in bold font. ^†^: mean ± standard deviation; BMI: body mass index; HFNC: high-flow nasal cannula oxygen therapy; ^‡^: GFR < 60 mL/min/1.73 m^2^; ^§^: long-term corticosteroid therapy, chemotherapy, or antirejection therapy in organ transplant recipients.

**Table 2 pathogens-12-00463-t002:** Clinical and mycological characteristics as well as treatments and outcomes of patients in the case group who developed candidaemia between 1 April and 30 November 2021 at AP-HM.

Patients	1	2	3	4	5
Demographic characteristics					
Gender	Male	Female	Male	Male	Female
Age (years)	62	68	46	72	69
Known immunosuppression	No	Anti-TNF α treatment	No	No	No
Comorbidities *	HBP	Diab, CV	Ob, Smo	HBP, Diab, Smo	HBP, Ob
Clinical characteristics					
Length of stay (days)	120	108	72	49	72
Mechanical ventilation	Yes	Yes	Yes	Yes	Yes
ECMO **	Yes	No	No	No	No
SARS-CoV-2	Yes	Yes	No	Yes	No
Abdominal surgery	No	Yes	Yes	No	No
Mycological culture					
Respiratory colonisation	*C. albicans*	*C. albicans*	*C. glabrata*	*C. glabrata*	*C. utilis*
Other colonised site	*Urine*	*Urine*	*No*	*No*	*Urine*
Blood cultures (BC)	*C. albicans*	*K. marxianus*	*C. glabrata*	*C. glabrata*	*C. metapsilosis*
No. of positive BCs	3	2	2	3	2
Treatments and outcome					
Antifungal treatments ***	Casp	Casp	Casp, Vorico	Casp	Casp, Fluco
Outcome	Alive	Alive	Alive	Alive	Alive

* HBP, high blood pressure; anti-TNF α, anti-Tumour Necrosis Factor alpha; Diab, diabetes; CV, cardiovascular disease; Ob, obesity; Smo, smoking. ** ECMO, extracorporeal membrane oxygenation.*** Casp, Caspofungin; Vorico, voriconazole; Fluco, fluconazole.

**Table 3 pathogens-12-00463-t003:** Immunomodulatory treatments used from 1 April to 30 November 2021 against SARS-CoV-2 infection in the case and control groups: BactC (patients with positive bacterial culture) and NegC (patients with negative respiratory sample cultures).

Characteristics	Cases = n (%)	BactC = n (%)	NegC = n (%)	*p*
SARS-CoV-2	53 (53)	29 (29)	17 (17)	<10^−4^
Intensive care unit	51 (51)	18 (18)	17 (17)	<10^−4^
Immunomodulatory treatments				
Dexamethasone	53 (53)	28 (28)	10 (10)	<10^−4^
Methylprednisolone	37 (37)	16 (16)	3 (3)	<10^−4^
Tocilizumab	6 (6)	7 (7)	0 (0)	0.6045

**Table 4 pathogens-12-00463-t004:** Multivariate unconditional logistical regression analysis of factors associated with *Candida* colonisation. Analyses compared the case group with both or either of the control groups BactC (patients with positive bacterial culture) and NegC (patients with negative respiratory sample cultures).

	Cases vs. BactC + NegC	Cases vs. BactC	Cases vs. NegC
	OR [95% CI] ***, *p*	OR [95%CI], *p*	OR, [95% CI], *p*
Oxacillin IV	4.42 [1.05–18.62], 0.0427	-	-
Ceftazidime	2.30 [1.13–4.66)], 0.0213	-	-
Cefepime	2.83 [1.42–1.64], 0.0030	17.66 [3.95–79.06], 0.0002	-
Piperacillin/tazobactam	1.99 [1.14–3.45], 0.0148	-	-
Linezolid	-	-	2.57 [1.23–5.36], 0.0120
Carbapenems	3.00 [1.58–5.68], 0.0007	22.84 [5.18–100.61], <10^−4^	
Length of stay	1.02 [1.01–1.03], <10^−4^	1.06 [1.02–1.09], 0.0007	1.02 [1.01–1.03], 0.0021
Diabetes	2.12 [1.10–4.07], 0.0248	7.44 [2.85–19.65], 0.0006	-
MV ^†^	3.48 [1.79–6.74], 0.0002	7.48 [2.85–19.65], <10^−4^	-
Fungal coinfection	6.71 [1.67–26.97], 0.0073	-	-
Solid malignancies	-	0.22 [0.06–0.73], 0.0138	-
Antifungal	-	-	4,17 [0.46–11.96], 0.0078

* OR [95%CI]: odds ratio and 95% confidence interval, ^†^ MV: mechanical ventilation.

## Data Availability

The data presented in this study are openly available in the IHU Méditerranée Infection site at https://doi.org/10.35081/0dm0-bj28, reference “COVID-19-Is-a-Confounder-of-Increased-Candida-Airway-Colonisation Dataset”.

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
