# Peer review of "COVID-19 Is a Confounder of Increased Candida Airway Colonisation"

_pathogens, 2023, doi:10.3390/pathogens12030463_

Round 1

Reviewer 1 Report

Froidefond and co-authors conducted a retrospective single-center study followed by a case-control study to analyze the prevalence of Candida spp. isolation from hospitalized patients during Covid, aiming to identify risk factors for yeast colonization.

Although the work is well written, there are few concerns that prevents me to endorse its publication. The matter discussed is not new in scientific discussion, as there are several other works already published and discussing the problem. In addition, I believe that the heterogeneity of the population analyzed in this retrospective study is an important limitation: maybe, authors should have analyzed the different population, with a special focus on the critically ill population. Another observation I would like to highlight is the absence of reference to Candida non-albicans spp. resistance to antifungals, which has been demonstrated to be on the rise and associated to Covid infection. 

Author Response

Froidefond and co-authors conducted a retrospective single-center study followed by a case-control study to analyze the prevalence of Candida spp. isolation from hospitalized patients during Covid, aiming to identify risk factors for yeast colonization.

Although the work is well written, there are few concerns that prevents me to endorse its publication.

  • We thank the reviewer for saying that this work is well written.

The matter discussed is not new in scientific discussion, as there are several other works already published and discussing the problem.

  • We agree with the reviewer that the research question is not new. Yet our study design was innovative. We find interesting to replicate studies in a scientific question, especially by using different methodological approaches and with data originating from various health care settings. Indeed, clinical practice habits and microbial ecology vary according to the geographical regions and patient management procedures. Moreover, a replicated result is less likely to be a false positive.

In addition, I believe that the heterogeneity of the population analyzed in this retrospective study is an important limitation: maybe, authors should have analyzed the different population, with a special focus on the critically ill population.

  • In contrast with the reviewer’s opinion on this subject, we believe that heterogeneity is rather a strength than a limitation of our study. Indeed our conclusions can be extrapolated to a much wider patient’s population than if we had selected a particular, narrower patients’ population.

  • In particular, focusing on the critically ill (like the majority of previous studies on this topic) would have allowed a more homogenous study population, but would have been less informative on studying the association between SARS-CoV-2 infection and Candida

Another observation I would like to highlight is the absence of reference to Candida non-albicans spp. resistance to antifungals, which has been demonstrated to be on the rise and associated to Covid infection. 

  • We agree with the reviewers’ remark, but this is another research question that we might address in another paper. In our University Hospital, we neither witnessed a notable increase in non-albicans Candida nor an increase in antifungal resistance. Our findings contrast with other published works conducted in different geographical areas. As replied above, we believe that these distinctive epidemiological features are worthy of being reported to the specialists and the scientific community.

Reviewer 2 Report

Froidefond and colleagues present an article entitled "Impact of COVID-19 on Candida airway colonization", in which they report results of a single-center study, where they analyze the prevalence of yeasts in respiratory samples and conduct a case-control study to determine the type of patients with airway colonization, the development of infection compared to a control group and the predisposing factors.  The study seems to be well designed and performed. The results are well presented, and the bibliography is adequate. 

Minor revision.

Line 261 in the table. In the column containing the variable analyzed there is n (%) which should be in the columns of cases and controls, to improve the presentation of the information. 

Author Response

Froidefond and colleagues present an article entitled "Impact of COVID-19 on Candida airway colonization", in which they report results of a single-center study, where they analyze the prevalence of yeasts in respiratory samples and conduct a case-control study to determine the type of patients with airway colonization, the development of infection compared to a control group and the predisposing factors.  The study seems to be well designed and performed. The results are well presented, and the bibliography is adequate. 

  • We thank the reviewer for this remark.

Minor revision.

Line 261 in the table. In the column containing the variable analyzed there is n (%) which should be in the columns of cases and controls, to improve the presentation of the information. 

  • We thank you for pointing this error out. The revised manuscript has been edited accordingly. Please see Table 3, page 15.

Reviewer 3 Report

General observations: The authors in the title declare to describe the impact of Covid -19 infection on Candida airway colonization. However reading the paper the impression is that the Covid-19 infection has just a minimal role in the entire manuscript. Probably the title should be different, focusing just on Candida colonization, as expressed in the aims, in the methods and in the results.  For example SARS-COV2 has just the last line in the table 1.

Globally the paper is well written, in good English, hwever  I feel that the paper is too long, too many data, tables and figures, not always crucial for the comprehension of the paper, for example figures 6,7,8 can be avoided, so is table 4.

My suggestion is to revise enterely the paper, reducing the number of figures and tables, and eventually changing the title.

Particular observations: In table 1 comorbidities, should be the "IMC" changed in BMI? 

Did the authors collect information about antifungal prohpylaxys in such at risk populatiom?

In the case-control study the authors compared the number of respiratory samples with positive bacterial culture, but they did not explain if more than one sample has been taken from the same patient in a short time lapse, so that the number of positive cultures does not give correct epidemiological information.

Author Response

General observations: The authors in the title declare to describe the impact of Covid -19 infection on Candida airway colonization. However reading the paper the impression is that the Covid-19 infection has just a minimal role in the entire manuscript. Probably the title should be different, focusing just on Candida colonization, as expressed in the aims, in the methods and in the results.  For example SARS-COV2 has just the last line in the table 1.

  • We thank the reviewer for his interest in our work, and for the time he spent for critically reviewing this manuscript.

  • We agree with him that the COVID-19 factor might seem diluted within the manuscript, in regard to the numerous factors we assessed in this study. We would like to stress that this work was triggered by our initial observation of a clear temporal relationship between the increase of yeast isolation in respiratory samples and the COVID-19 pandemic (e.g. notably in March and April 2020, the last quarter of 2020, from February to May 2021, in August and September 2021, and from December 2021 to January 2022). Thereafter, we aimed to assess whether SARS-CoV-2 infection and/or several other risk factors were associated with Candida

  • Because of the way in which this study was designed and conducted, we believe legitimate to maintain the title as submitted. But we do agree with the reviewer that SARS-CoV-2 infection acted as a confounding factor, as we pointed out in our conclusion.

Globally the paper is well written, in good English, hwever  I feel that the paper is too long, too many data, tables and figures, not always crucial for the comprehension of the paper, for example figures 6,7,8 can be avoided, so is table 4. My suggestion is to revise enterely the paper, reducing the number of figures and tables, and eventually changing the title.

  • We agree with the reviewer’s remarks and transferred figures 6, 7, and 8, and also Table 4 in the supplementary material. These have been re-named and are cited in the manuscript accordingly to author’s guidelines. Please see Supplementary Material, Figures S1, S2, S3 and Table S1.

Particular observations: In table 1 comorbidities, should be the "IMC" changed in BMI? 

  • We thank you for pointing this out mistake. The revised manuscript has been edited accordingly. Please see Table 1, page 10.

Did the authors collect information about antifungal prohpylaxys in such at risk populatiom?

  • This is a fair point. As mentioned in the manuscript, all patients with a diagnosis of IFD received systemic antifungal therapy, and also several patients from the Case, Tbact and Tneg groups received antifungal therapy, either after documenting colonization or as probabilistic treatment of a possible IFD.

  • Patients who received antifungal prophylaxis (or preemptive treatment) following colonization assessment received caspofungin (Cases = 3 patients), fluconazole (Cases = 1 patient) or liposomal amphotericin B (Tneg = 1 patient).

  • Patients who received probabilistic treatment received caspofungin (Cases = 3 patients ; Tbact = 1 patient ; Tneg = 1 patient), fluconazole (Cases = 1 patient) and liposomal amphotericin B (Cases = 1 patient).

  • Although recommended by several experts and scientific societies, administration of prophylactic antifungal therapy in high-risk patients depends much on the managing clinician/team’s working habits and infection management strategies. We generally recommend to assess Candida colonization in high-risk patients through culture-based Candida colonization index. In case of colonization (documented by culture isolation of Candida from at least 3 out of 5 anatomical regions sampled), we suggest initiation of pre-emptive treatment with relevant antifungals, in regard to the yeast isolate’s species and susceptibility profile.

In the case-control study the authors compared the number of respiratory samples with positive bacterial culture, but they did not explain if more than one sample has been taken from the same patient in a short time lapse, so that the number of positive cultures does not give correct epidemiological information.

In the Cases and Control definitions, we specified that the patients included must have had at least three respiratory samples on separate days to assess Candida colonization prior being included. Yet, each patient was counted once, and analyzes were based on patients, and not samples.

Round 2

Reviewer 1 Report

I thank the authors for the clarifications.

I believe the work is ready to be accepted in its present form. 

Author Response

I thank the authors for the clarifications.

I believe the work is ready to be accepted in its present form. 

We thank you for your interest in this work and for the time taken to review this manuscript.

Reviewer 3 Report

The manuscript has been revised according to some points of the review report. In particular, I still feel a mismatch between the title and the text, and no correction has been done to mitigate the discrepancy. For example the aims in the abstract might report the same sentence used in the introduction: This work aimed to assess the association of risk factors, particularly SARS-CoV-2 infection, with Candida respiratory tract colonization

I have no more comments regarding the other points 

Author Response

The manuscript has been revised according to some points of the review report. In particular, I still feel a mismatch between the title and the text, and no correction has been done to mitigate the discrepancy. For example the aims in the abstract might report the same sentence used in the introduction: This work aimed to assess the association of risk factors, particularly SARS-CoV-2 infection, with Candida respiratory tract colonization

I have no more comments regarding the other points 

We thank the reviewer for the time taken to review this manuscript. As to be in better accordance with the reviewer's suggestions, we edited the title, and propose "COVID-19 is a confounder of increased Candida airway colonization" (please see the revised manuscript's title).

Moreover, we also edited the abstract as suggested (please see abstract, page 1, lines 3-4 "The present study aimed to test the impact of several factors on Candida airway colonization, including SARS-CoV-2 infection."

We hope that these modifications will satisfactorily clarify the work's aims, objectives and methods, and thank again the reviewer for the time spent critically reviewing this manuscript.